# Tying Small Changes to Large Outcomes: The Cautious Promise in Incorporating the Microbiome into Immunotherapy

**DOI:** 10.3390/ijms22157900

**Published:** 2021-07-23

**Authors:** Justin Chau, Jun Zhang

**Affiliations:** 1Division of Hematology, Oncology and Blood & Marrow Transplantation, University of Iowa Hospitals and Clinics, Iowa City, IA 52246, USA; justin-chau@uiowa.edu; 2Division of Medical Oncology, Department of Internal Medicine, University of Kansas Medical Center, Kansas City, KS 66160, USA; 3Department of Cancer Biology, University of Kansas Cancer Center, Kansas City, KS 66160, USA

**Keywords:** microbiome, immunotherapy, immune checkpoint, toxicity, adverse events, lung cancer

## Abstract

The role of the microbiome in immunology is a rapidly burgeoning topic of study. Given the increasing use of immune checkpoint inhibitor (ICI) therapy in cancers, along with the recognition that carcinogenesis has been linked to dysregulations of the immune system, much attention is now directed at potentiation of ICI efficacy, as well as minimizing the incidence of treatment-associated immune-related adverse events (irAEs). We provide an overview of the major research establishing links between the microbiome to tumorigenesis, chemotherapy and radiation potentiation, and ICI efficacy and irAE development.

## 1. Introduction

As indications for immune checkpoint inhibitors (ICI) for treatment of malignancies have broadened, outcomes remain variable and difficult to predict. Save for specific tumor traits such as genomic microsatellite instability (MSI-H), high programmed cell death ligand-1 (PD-L1) expression, or other genetically hypermutated (POLε) states, it appears that only a minority of patients respond to ICI therapy. Despite low response rates, patients whose cancers regress under ICI therapy may be more likely to enjoy longer progression-free survival. Unfortunately, this suboptimal chance of benefit is accompanied by increased risk of immune-related adverse effects (irAEs). These phenomena are associated with ICI activation of T cells throughout the body, potentially leading to a multitude of occasionally life-threatening autoinflammatory organopathies. In short, the current state of immunotherapy in oncology leaves much room for improvement.

The microbiome has emerged as a possible therapeutic avenue as well as conduit for understanding mechanisms by which ICI effects are modulated: why do some patients develop crippling toxicities from treatment, while others virtually none? How can we explain the variance in outcomes between patients with similar cancers and tumor PD-L1 expression following treatment with the same drug? Is the oft-noted association between development of irAEs with response to ICI therapy in actuality describing two phenomena that can be decoupled?

The microbiome has already been recognized for its significant role in human metabolism. Specific bacteria in the gut are responsible for digestion and metabolism in a symbiotic manner. There is increasing study of the microbiome’s role in gluconeogenesis and lipid metabolism [1,2]. The gut microbiome also affects the metabolism of medications, utilizing a wide variety of enzymatic actions including but not limited to reduction, decarboxylation, demethylation, deamination, and hydrolysis of prodrugs [3]. It is not simply the presence of the microbiome that mitigates these outcomes—specific bacterial genera are responsible for critical functions. Lastly, the microbiome is increasingly associated with disease states, including neuropsychiatric [4,5] and inflammatory bowel diseases [6,7], and, as we will discuss, cancer.

On a daily basis, our microbiome—and the metabolomic networks formed within it—are instrumental in mitigating the body’s inflammatory state [8,9], but its composition can be affected by external insults [10]. Using this framework, we can begin to appreciate the search for links connecting the microbiome, cancer, and the efficacy and toxicity of immunotherapy, its possible treatment.

## 2. The Microbiome and Cancer

It has been long theorized that the microbiome plays a pivotal role in carcinogenesis, although the nature of its exact contribution is yet to be confirmed. Multiple viruses, including hepatitis B and C, HTLV-1, and HPV [11], have been associated with cancer development. Studies in the 1970s and 1980s found that germ-free mice were less likely to develop spontaneous cancers compared to conventional mice [12,13]. This review will focus on bacterial etiologies for carcinogenesis. The evidence linking the two encompasses multiple tumor types and sites of origin. Here, we review evidence that the microbiome has contributed to—or directly resulted in—cancer development. We then review the possible mechanisms involved in this phenomenon.

### 2.1. Alterations in Microbiome Composition Are Associated with Carcinogenesis

The breadth of findings tying microbiome composition to carcinogenesis vary widely. They can be very narrow, attributing one or two bacterial genera to tumor development; other studies implicate alterations in the abundance of an entire phylum or even the overall number of detectable taxonomic species in a microbiome. Studies of colorectal cancer (CRC) patients have associated decreased α-diversity, or overall species enrichment, of the gut microbiome with tumorigenesis [14,15]. At the phylum level, decreased relative abundance of *Firmicutes* has been associated with CRC [14]. More granular findings include association of *Bacteroides*, *Coprococcus*, *Corynebacterium*, *Enterococcus*, and *Neisseria* enrichment with higher likelihood of CRC development, although the causative mechanisms remain unclear [16,17]. It is possible that with more data we eventually identify specific bacterial functional networks, or metabolomes, associated with these clinical outcomes.

In our and others’ research, lung cancer patients had different gut and respiratory microbiome compositions compared to their healthy controls [18,19,20]. Zhuang et al. found reduced enrichment of *Actinobacteria* in the gut microbiome of lung cancer patients [18]. In a multirepository study of the respiratory microbiome of lung cancer patients, an overabundance of *Proteobacteria* and decrease in *Firmicutes* was identified [19]. Other changes in the composition of the microbiota attributed to lung cancer, such as increased *Fusobacteria* and *Bacteroidetes* in the gut microbiota of lung cancer patients [14], have been reported but yet to be reliably replicated. These findings have bolstered the concept of immune cross-talk—associations or coordination between disparate microbiome sites in mediating disease states. This phenomenon has been described in various autoimmune diseases such as asthma and COPD [21,22,23] but thus far has not been established in cancer.

### 2.2. The Potential Link between Microbiota-Induced Environmental Changes and Carcinogenesis

Multiple mechanisms have been posited to explain the significance of observed dysbiosis. *Fusobacterium* has been associated with development of CRC but has also been identified inhabiting hepatic metastatic lesions [16,24]. This implies that *Fusobacterium* can facilitate tumor growth and metastasis, either by generating a tumor microenvironment tolerant of its development or by directly assisting with tumor cell adhesion [25,26].

Significant preclinical evidence has identified bacteria-induced localized inflammation as a potential common thread. The colonization of the stomach with the *Proteobacterium Helicobacter pylori* has been shown to lead to gastric activation of NOD1-like receptor, which in turn stimulates an IL-33 dependent response, leading to generation of inflammatory macrophages and T-cell phenotypes [27,28,29]. Similarly, chronic inflammatory changes from *Salmonella typhi* and other *Enterococcus* genera have also been associated with development of biliary cancer [30,31].

A major 2019 study by Jin et al. described a novel mechanism in the development and growth of lung cancer. Intratracheal transfection of cre recombinase encoded adenovirus in mice caused activation of parenchymal cell *Kras^G12D^* mutation and *Tp53* deletion, leading to tumorigenesis. Local respiratory bacteria induced MYD88-dependent lung inflammation and led to activation of IL-17-producing T cells, facilitating cancer development. Notably, mice whose commensal respiratory bacterial loads were ablated exhibited slowed progression of disease. This further reinforces the theory that alterations in the local/respiratory microbiome influences the manner of carcinogenesis [32]. Consistent with this, Tsay et al. found higher prevalence of lower airway dysbiosis in advanced lung cancer patients associated with poor prognosis [33].

Multiple other studies corroborate adjustments in the tumor microenvironment to induce an inflammatory state. In addition to its effects in the gastric body, *H. pylori* colonization in the mouse intestinal tract has been shown to increase production of c-myc, interleukin-1 (IL-1), IL-6, and tumor necrosis factor (TNF), but reduced interferon-gamma (IFN-γ), with subsequent development of colorectal cancer [34,35]. *Bacteroides fragilis* generates enterotoxins causing colitis and subsequent production of IL-17A via recruitment of Th17 and γδ T cells, among others [36]. The inflammation appears to be associated with development of colorectal cancer in both humans and mice [36,37].

The contribution of the microbiome in generating or dampening inflammatory immune responses, predominantly via T cell population shifts, has drawn interest for both pathophysiologic and therapeutic reasons. The bedrock principle of ICI efficacy is to harness cytotoxic and effector T cells to generate an anti-tumor effect—it is this modulatory aspect of the microbiome, carcinogenic it may be, that may also provide a way to improve its treatment.

## 3. The Role of the Microbiome in Chemotherapy and Radiation Therapy

Less nebulous than its role in carcinogenesis is the clear fact that specific members of the microbiome are responsible for metabolizing—thus potentiating or weakening—particular drugs or therapies. This is valuable for demonstrating how the microbiome can be used to affect the immune system, even if the intervention was not originally intended to induce an immune response.

An example of the direct activity bacteria can have on chemotherapy was demonstrated by Geller et al., showing the intracellular effects of gemcitabine can be abrogated by a deamination reaction initiated by *Mycoplasma hyorhinis*—an abrogation repotentiated by depleting intratumoral bacteria [38]. Similarly, mouse models demonstrated the presence of *Parabacteroides disastonis* in the gut can also decrease the efficacy of doxorubicin [39,40].

The adverse effects of chemotherapies can also be mitigated by bacteria. A well-known adverse effect of irinotecan, mucositis, and diarrhea is one major example. The active metabolite of irinotecan, SN-38, is glucuronidated (inactivated) in the liver, and then excreted fecally. However, upon coming into contact with gut bacteria, the glucuronidate moiety is removed and the drug is reactivated in the gut, leading to enterospecific toxicities [41].

Interestingly, the interactions between microbiota and medications are not always dependent on enzymes: effects can also be secondary to the intended therapeutic mechanism. In mice, cyclophosphamide was shown to induce interferon-γ and IL-17, a phenomenon occurring to a lesser degree in germ-free counterparts [39]. Further studies showed that in addition to its DNA cross-linking effect, cyclophosphamide disrupts the enteric lumen, leading to translocation and antigenic presentation of specific gut microbiota (*Enterococcus hirae, Lactobacillus johnsonii*, and *Barnesiella intestihominis*) in the mesenteric lymph nodes, with resultant stimulation of an effector immune response [39,42] that augments its principal antineoplastic function.

In addition to its previously mentioned contribution to colorectal cancer metastasis, colon cancer patients with higher abundances of *Fusobacterium* in their gut microbiota were found to be less likely to respond to 5-fluorouracil and oxaliplatin, the principal components of first-line therapy for metastatic colorectal cancer. This was theorized to occur by *F. nucleatum*-specific activation of TLR4 and MYD88 pathways, leading to tumor cell autophagy in response to chemotherapy rather than apoptosis [43,44].

Alterations in the microbiome have also been seen after treatment with radiation therapy. In a similar fashion to cyclophosphamide, researchers found that total body irradiation (TBI) was able to improve the antitumor efficacy of CD8+ T cells in mice by upregulating influx of circulating inflammatory cytokines via increased enteric lumen permeability [45]. In another study, *Proteobacteria* exhibited increased abundance, and *Firmicutes* decreased abundance, in mouse models that developed radiation proctitis. That inflammation was tied to postradiation dysbiosis, whose metabolic functions generated inflammatory cytokine IL-1β [46].

It is clear that the gut microbiome is instrumental in mediating the effects and toxicities associated with multiple treatment modalities, and thus begs the question of its contribution to the same in immune checkpoint blockade.

## 4. The Microbiome in Immunotherapy

### 4.1. Immunotherapy Efficacy

To date, several studies regarding the gut microbiome and immunotherapy efficacy have been performed [15,20,47,48,49,50,51,52]. The majority of initial studies focused on patients treated for melanoma, a disease with high propensity for response to immunotherapy; this has since broadened to include multiple tumor types. Though these studies have also utilized metagenomic shotgun sequencing, a more expensive technique that can provide metabolic pathway information not available with standard 16S rRNA sequencing, they have largely associated these outcomes with changes in the gut microbiome alone.

At a superficial level, the microbiome has already been associated with immunotherapy efficacy [53,54]. Higher overall alpha-diversity, or overall number of identifiable bacterial species, has been associated with improved likelihood of response to immunotherapy in multiple studies [15,55]. Conversely, patients with melanoma treated with antibiotics within 30 days of ICI initiation were less likely to respond to immunotherapy, as well as more likely to have faster progression of disease [47,56]. Critically, this ability of a diverse microbiome to regulate ICI response appeared to be transferable—in mouse models, fecal transplantation from immunotherapy responders into previously nonresponding mice was shown to repotentiate efficacy [15,57]. This hints that a diverse microbiome may be needed to obtain clinical benefit from immunotherapy, and that these outcomes can be modulated.

Immunotherapy efficacy has also been attributed to the relative enrichment (or lack thereof) of specific species in the gut microbiome. In melanoma patients, increased abundance of *Bacteroides* species were associated with poor clinical benefit; higher abundances of *Faecalibacterium*, *Holdemania*, and other *Firmicutes* bacteria in the gut were associated with higher odds of long-term benefit [48,50,58]. Other bacteria associated with worsened likelihood of ICI efficacy include members of the *Proteobacteria* phylum [58].

Other studies have linked ICI outcomes in nonmelanomatous cancers to gut microbiome composition. We found a similar association between *Firmicutes* and ICI response in our 16S evaluation of lung cancer patients [20]; these findings were echoed in a separate cohort that had undergone whole genome shotgun sequencing [48]. Routy et al. found that *Akkermansia*, a member of the phylum *Verrucomicrobia*, was enriched in NSCLC patients who responded to ICI. When broadening that cohort to include renal cell carcinoma patients—for which ICI typically comprises first-line therapy with good propensity for response—*Akkermansia muciniphila* was still associated with best response to therapy, as well as increased duration of response [47]. This is remarkable for the identification of microbiome alterations that may reflect immunotherapy response across cancer types.

More recently, the microbiome’s contribution to ICI efficacy in gastrointestinal cancers has come into focus. A study of 89 patients with advanced gastrointestinal cancer (colorectal, esophageal, gastric, etc.) treated with ICI whose fecal samples underwent 16S sequencing were found to have higher abundances of *Ruminococcaceae* and *Lachnospiraceae* in patients with prolonged progression-free survival, and high levels of *Bacteroides* species in those with shorter progression-free survival [59]. These are similar to the findings seen in the earlier melanoma studies from Routy and Gopalakrishnan, implicating specific *Firmicutes* phylum bacteria in association with improved outcomes. The DELIVER trial evaluated differences in the gut microbiome of advanced gastric cancer patients treated with nivolumab immunotherapy. With extremely stringent (Bonferroni corrected) analysis, changes in *Odoribacter* (phylum *Bacteroidetes*) and *Veillonella* (phylum *Firmicutes*) were found to be associated with improved responses to immunotherapy [60]. In concordance with these findings, our recent systematic review observed that across various solid tumors, patients who had enriched abundance in bacterial *Firmicutes* and *Verrucomicrobia* phyla almost universally had better response from ICIs, whereas those who were enriched in *Proteobacteria* universally presented with unfavorable outcome [58].

What could be the mechanism(s) linking the microbiome to immunotherapy response? Multiple possible mechanisms are outlined in Figure 1. One theory is that specific bacterial antigens stimulate T cell populations that then, primed, cross-react against tumor neoantigens. *Bifidobacterium* has been demonstrated to be associated with increased likelihood of ICI response [49]. A striking study by Bessell et al. identified an epitope in the *Bifidobacterium breve* genome (SVY) whose homology to an SIY model tumor antigen led to development of SVY-specific T cells. In vivo, these SVY cells targeted SIY-positive melanoma cells and subsequently prevented tumor growth [61]. The transfer of commensal fecal bacteria from mice who were known to have slow melanoma growth, attributed to increased relative abundance of *Bifidobacterium*, into mice with faster-growing melanoma led to a similarly decreased rate of tumor growth and increased antigen-specific T cell tumor infiltration in the latter population [49].

As mentioned in the microbiome and cancer section, antigenic homology is not the only way bacteria can elicit a T cell response. Bacterial toxins can also stimulate recruitment of T cells, which in turn release inflammatory cytokines such as IL-17 [28], and counters the adaptive immune tolerance provided by regulatory cytokines such as IL-10. This broadly inflammatory myelopoietic function could potentiate immunotherapy efficacy. Such outcomes might be made possible via multiple avenues, such as activation of NOD-like receptors, toll-like receptors (TLRs), or G protein-coupled receptors (GPCRs), some of which are currently under study in early phase clinical trials. However, extensive study of the specific bacteria associated with improved efficacy outcomes may shed further light on uninvestigated mechanisms. Possible leads include the observed conversion of a subset of Foxp3+ regulatory T cells to pro-inflammatory helper Th17 cells [62] or decreasing the suppressive activity of existing regulatory T cells [63]. Another emerging target for modulation is the γδ T cell population, which appears to facilitate a complex variety of cytotoxic and antibody-mediated anti-tumor pathways, including IFN-γ, IL-10/IL-4, local recruitment of monocytes and neutrophils, and antigen presentation, with more specificity than typical effector T cells [36,64]. Despite the promising use of these cells to control malignant disease, their population relative to the overall T cell population is small [64]. Determining which bacteria may be able to stimulate an increase in this population may improve efficacy outcomes. An unexpected finding in a study by Benakis et al. not only demonstrated modulation of the γδ T cell population by the intestinal microbiome but was also able to link the increase in γδ T cells to development of inflammatory changes in distant organ sites (ischemic stroke) [65].

Another theory involves specific metabolic functions of the bacterial genera implicated. In general, the *Bacteroidetes* phylum is associated with poorer odds of ICI efficacy, but for reasons that remain unclear, two of its members, *Bacteroidetes thetaiotaomicron* and *Bacteroides fragilis*, mark possible significant exceptions to the rule [15,58]. These phenotypes are characterized by outer membrane vesicles that also contain proinflammatory molecules such as lipopolysaccharide (LPS) [8]. Repletion of these specific genera in bacteria-depleted mice improved response to anti-CTLA-4 therapy when previously there was none [51]. *B. fragilis* has been shown to produce polysaccharide A (PsA), which also stimulates regulatory T cell production [66]. This subtle but important intraphylum distinction in bacterial function may become critical to understanding the pathways utilized in ICI potentiation.

A common question is whether these differences in composition actually influence immunotherapy efficacy, as opposed to reflecting its byproduct. The latter possibility is suggested by the observation that injection of mice with anti-CTLA-4 therapy induces a fecal decrease in *Bacteroidales* and increase in small bowel concentrations of *B. thetaiotaomicron.* However, feeding germ-free mice mixtures of the same *Bacteroides* species reconstituted the anti-CTLA-4 effect, rendering the answer to this question murky at best [51,67]. This is an area necessitating further study, especially given the fact that transient changes to the microbiome can occur following environmental stimuli.

As previously seen, much attention has been paid to changes in the gut microbiome, but given the finding by Jin et al. with respect to local bacterial factors in pulmonary carcinogenesis [32], it seems prudent to explore the contribution of the respiratory microbiome to ICI efficacy in lung cancer, and a possible correlation between lung and gut microbiomes. We are probably the first to address this question by analyzing the nasal and buccal microbiome given its proximity to the lungs: we observed that enrichment of *Finegoldia* (of phylum *Firmicutes*) in nasal and *Megasphaera* (also of phylum *Firmicutes*) in buccal swabs correlated with better response in NSCLC patients treated with chemoimmunotherapy [20].

### 4.2. Immune-Mediated Toxicities

One of the most complex issues facing the use of immunotherapy is that while ICIs are largely better tolerated than chemotherapy, they also carry the possibility of developing an immune-related adverse event, with clinical severities ranging from asymptomatic to fatal. As immunotherapy mobilizes immune cells across the body, the possible adverse effects a patient may develop can include colitis, dermatitis, nephritis, hepatitis, neuritis, pneumonitis, myocarditis, and even endocrinopathies [68]. With exception of patients with preexisting autoimmune disease, which may inappropriately flare with ICI use, it is still unclear which patients will ultimately develop toxicities and why. Given the gut microbiota’s capacity to mitigate chemotherapy effects, further study is being directed toward its potential role in irAEs.

The fecal microbiota of melanoma patients who developed colitis after being treated with ipilimumab, an anti-CTLA-4 immune checkpoint inhibitor, were found to have decreased abundance of *Firmicutes* at onset of colitis, but enriched *Firmicutes* at baseline, whereas increased baseline *Bacteroidetes* was seen in patients who did not develop colitis [50]. Our systematic review noted these same trends across various solid tumor types receiving ICIs [58]. In addition to mirroring these findings, our research also found that increased *Proteobacteria* are similarly associated with decreased likelihood of irAE development [20].

These phenomena may be due to the metabolic products of these bacteria. When considering the negative association of irAEs with *Bacteroidetes* species, *B. fragilis* has been demonstrated to suppress proinflammatory cytokine production through secretion of polysaccharide A [69]. *Firmicutes* bacteria are responsible for production of short-chain fatty acids (SCFAs), metabolites that exert regulatory effects on inflammation and T cell differentiation. While a recent clinical study demonstrated that elevated fecal SCFA concentration is significantly correlated with better clinical outcomes with anti-PD-1 treatment [70], prior studies also showed that SCFAs could increase the level of IL-17, a pro-inflammatory cytokine that plays a critical role in irAEs [71,72]. These findings are consistent with the clinical observation that patients who experience greater irAEs tend to have better response from ICIs [73]. However, SCFAs were also found capable of inducing regulatory T cells and thus an anti-inflammatory response [74,75]. In addition, pre-treating dendritic cells with butyrate (a 4-carbon SCFA) led to decreased generation of inflammatory cytokines [74,75]. This conflicting information shows that SCFAs (and probably metabolites in general) cannot fully explain the association of *Firmicutes* with the development of irAEs, and a thorough understanding of their role in various immunologic scenarios are needed. Nevertheless, these observations beg the question whether the therapeutic response and irAEs from ICIs can be decoupled.

The 2015 Vetizou study found that repletion of *B. fragilis* and *B. cepacia* improved CTLA-4 ICI efficacy in germ-free animal models and simultaneously reduced the risk of developing ICI-induced colitis [51]. *Bifidobacterium* has also been reported to hamper development of ICI-mediated colitis [76]. Attempts to use an appropriate mix of *Firmicutes* and *Bacteroidetes* to enhance immunotherapy response yet mitigate irAEs were also found to have promising therapeutic implications [77]. All these findings implicate the possibility of improving ICI efficacy while simultaneously maintaining its tolerability—a boon that could improve patients’ clinical experience across all cancer types.

## 5. Uncertainty

The increased interest in this field has uncovered a number of additional questions. Although overall trends in alpha-diversity with respect to progression-free and overall survival appear to be concordant, the specific genera implicated in each study are not always congruent. Further, the 16S and metagenomics tools/algorithms used to study these populations are not standardized. This disparity in cross-study comparisons has led to the concern that the lack of replicability may indicate the findings in these studies are more noise than signal. It is important to note, then, that the composition of the microbiome can vary widely from geographic environ to environ [78,79]. Further, when comparing implicated bacteria at a higher taxonomic level, the findings become more aligned: an association of *Firmicutes* bacteria with increased likelihood of ICI response and propensity for irAEs, and increased enrichment of *Bacteroidetes* and *Proteobacteria* in ICI nonresponders as well as those less likely to suffer irAEs.

It is also important to stress that a bacteria’s taxonomic branch does not necessarily dictate its function: the observed enrichment of specific *Bacteroidetes* species in the gut microbiome of ICI responders bucks the trend exhibited by the rest of the phylum [51]. Thus, the more important focal points here may not necessarily be the specific bacteria enriched, but the functions they perform. Metagenomic studies provide descriptive criteria of metabolic pathways that taxonomic 16S studies cannot; with improved standardization of raw data collection, the ability to combine and improve these studies’ discriminatory power may also increase.

## 6. Conclusions

This review has outlined outcomes in studies involving skin, gastric, lung, kidney, and colon cancer patients. It can be argued that specific associations between microbiome composition and the metabolism of immunotherapy are likely to be independent of disease subtype, couched by the possibility there may be some cancers that will never respond to immunotherapy regardless of its potentiation.

However, the implications of these microbiome studies are important on several levels. In the most rudimentary sense, the identification of a microbiome composition favorable to immunotherapy efficacy or tolerability could be leveraged simply by using fecal transplantation to achieve the desired makeup. Yet the possible benefits extend far beyond that: they beget a better understanding of the mechanisms, proteomic and epigenetic, that may be involved in modulating immunotherapy responses in vivo. This field remains in its infancy, but its value is beginning to make itself apparent in preclinical studies for other treatments. For example, mouse models treated with a glucuronidase inhibitor have been shown to dampen irinotecan enterotoxicity [80], providing a potential pharmacologic intervention that could allow more patients to tolerate longer exposure to an efficacious drug. Multiple international studies are currently underway in an attempt to further characterize this relationship—Table 1 provides an abridged list of current immunotherapy-microbiome trials in development.

The true value of microbiome study likely lies in the use of pathway and network analysis to elucidate complex mechanisms leading to tumor growth and immunotherapy potentiation. If the microbiome truly represents an independent factor in immunotherapy outcomes, it would comprise another tumor-agnostic therapeutic avenue in an ever-expanding repertoire, a major development for oncology patients who have not yet benefited much from ICI use. This is why continued work in this vein remains vital to the whole of clinical oncology.

## Figures and Tables

**Figure 1 ijms-22-07900-f001:**
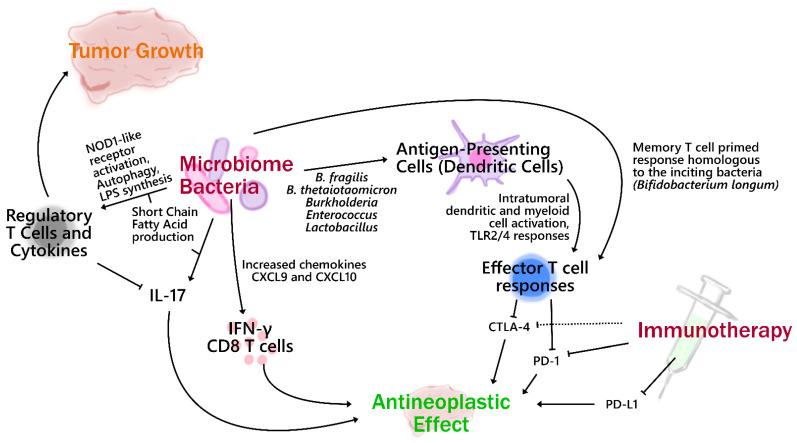
Possible mechanisms utilized by the microbiome in immunotherapy modulation. The microbiome is a complex environment in which multiple mechanisms of action are not entirely understood. Certain behaviors such as production of inflammatory cytokines or stimulation of regulatory T cells are often not entirely straightforward and may differ not just from phylum to phylum but between genera within a phylum. This diagram illustrates only a few of the possible mechanisms that have been demonstrated thus far in preclinical and clinical studies. Lines ending in perpendicular heads, whether dotted or unbroken, indicate inhibitory relationships.

**Table 1 ijms-22-07900-t001:** Active studies examining microbiome and immunotherapy outcomes.

ClinicalTrials Number	Trial Title	Primary Location	Study Type	Site Samples	Description/Techniques	Cancer Type	Status
NCT04107168	Microbiome Immunotherapy Toxicity and Response Evaluation	Cambridge, UK	Observational cohort study	Gut Oral	Attributions of microbiome composition to immunotherapy efficacy and toxicity	Melanoma Renal Cancer Lung Cancer	Recruiting
**NCT04636775**	**Microbiome in Immunotherapy-naïve NSCLC Patients Receiving PD-1/L1 Blockade**	**Kansas City, KS, USA**	**Observational cohort** **study**	**Gut** **Nasal** **Oral**	**PD-L1 expression16S rRNA** **Metagenomic sequencing**	**Lung cancer**	**Recruiting**
NCT03643289	Predicting Response to Immunotherapy for Melanoma with Gut Microbiome and Metabolomics	Middlesex, London and Manchester, UK	Observational cohort study	Gut Peripheral blood	Metagenomic sequencing Peripheral blood monocytes	Melanoma	Recruiting
NCT04579978	Tumor Immunotherapy and Microbiome Analysis	ON, Canada	Observational cohort study	Gut Peripheral blood	16S rRNA (stool samples) Metagenomic sequencing IgA sequencing Flow cytometry at progressive disease	All solid tumors	Recruiting
NCT03686202	Feasibility Study of Microbial Ecosystem Therapeutics (MET-4) to Evaluate Effects of Fecal Microbiome in Patients on Immunotherapy	ON, Canada	Randomized open-label clinical trial	Gut	MET-4, novel transplantation of live bacterial cultures from a healthy donor	All solid tumors	Recruiting
NCT02960282	Gut Microbiome in Fecal Samples from Patients with Metastatic Cancer Undergoing Chemotherapy or Immunotherapy	CA, USA	Observational cohort study	Gut	16S rRNA (stool samples) Metagenomic sequencing Meta-transcriptomics analysis Meta-proteomics analysis	Metastatic carcinoma	Recruiting
NCT04711330	Response and Toxicity Prediction by Microbiome Analysis after Concurrent Chemoradiotherapy	Vlaanderen, Belgium	Observational cohort study	Oropharyngeal Gut	16S rRNA (throat and stool samples)	Non small cell lung cancer	Not yet recruiting
NCT04638751	ARGONAUT: Stool and Blood Sample Bank for Cancer Patients	CA, USA	Observational cohort study	Gut Peripheral blood	Not yet determined	Non small cell lung cancer Colorectal Cancer Triple Negative Breast Cancer Pancreas Cancer	Recruiting
NCT04645680	Effect of Diet on the Immune System in Patients with Stage III-IV Melanoma Receiving Immunotherapy, DIET Study	Houston, TX, USA	Randomized double-blinded clinical trial	Gut	Isocaloric high-fiber diet vs isocaloric control diet	Melanoma	Recruiting
NCT03817125	Melanoma Checkpoint and Gut Microbiome Alteration with Microbiome Intervention	Los Angeles, CA, USA Boston, MA, USA	Randomized blinded clinical trial	Gut	Placebo or antibiotic in combination with nivolumab, followed by possible microbiome intervention	Melanoma	Active, not recruiting
NCT03772899	Fecal Microbial Transplantation in Combination with Immunotherapy in Melanoma Patients (MIMic)	ON, Canada QC, Canada	Open label clinical trial	Gut	Fecal microbiota transplantation	Melanoma	Recruiting
NCT04264975	Utilization of Microbiome as Biomarkers and Therapeutics in Immuno-Oncology	Seoul, Korea	Open label clinical trial	Gut	Fecal microbiota transplantation	All solid tumors	Recruiting
NCT04163289	Preventing Toxicity with Renal Cancer Patients Treated with Immunotherapy Using Fecal Microbiota Transplantation (PERFORM)	ON, Canada	Open label clinical trial	Gut	Fecal microbiota transplantation	Renal Cell Carcinoma	Recruiting
NCT04054908	Gut Microbiome in Colorectal Cancer	San Francisco, CA, USA	Observational cohort study	Gut	16S rRNA	Colorectal Cancer	Recruiting
NCT04567446	Discovery of Microbiome-based Biomarkers for Patients with Cancer Using Metagenomic Approach	Val De Marne, France	Observational cohort study	Gut Blood Oral	Metagenomic sequencing	All solid tumors	Recruiting
NCT04552418	Intestinal Microbiome Modification with Resistant Starch in Patients Treated with Dual Immune Checkpoint Inhibitors	Ann Arbor, MI, USA	Open label clinical trial	Gut	Potato starch Metagenomic sequencing	All solid tumors	Not yet recruiting
NCT04056026	A Single Dose FMT Infusion as an Adjunct to Keytruda for Metastatic Mesothelioma	Ventura, CA, USA	Open label clinical trial	Gut	Fecal microbiota transplantation	Mesothelioma	Completed
**NCT04680377**	**Using Microbiome to Predict Durvalumab Toxicity in Post- Concurrent Chemoradiation Therapy (CCRT) NSCLC Patients**	**Kansas City, KS, USA**	**Observational cohort study**	**Gut** **Nasal** **Oral**	**Metagenomic sequencing**	**Lung cancer**	**Recruiting**

Bolded studies indicate projects by the authors of this manuscript.

## Data Availability

Not applicable.

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
