# Peer review of "Tying Small Changes to Large Outcomes: The Cautious Promise in Incorporating the Microbiome into Immunotherapy"

_ijms, 2021, doi:10.3390/ijms22157900_

Round 1

Reviewer 1 Report

The authors summarize studies that show that the microbiota is related to caner pathogenesis and outcome to ICI therapy.

These are to date two well known and documented aspects .

What is an exciting and novel field  required for improving immunotherapy in patients with cancer is to understand how microbiota-host interactions fuel chronic inflammatory responses and promote pathogenic immune cells and cancer progression.

The authors mention briefly the induction of Treg and antigen-cross presentation to T cells.

However, it is becoming clear that microbiota-dependent signals sustain chronic TLR-driven inflammation and immunopathology by promoting inflammatory myelopoiesis. These data implicate microbiota as a central pathogenic contributor to chronic inflammation, and provide rationale for future attempts to target microbiota-dependent regulation of inflammatory myelopoiesis for therapeutic benefit in chronic inflammation driven cancer .

The authors need to describe and discuss in depth the pathogenic mechanisms of microbiota-dependent signals regulating inflammation in cancer, especially key immune responses as inflammatory myelopoiesis that have a critical role in the outcome of the anti-tumor effector T cell response.

Author Response:

We very much appreciate this comment and absolutely agree that the study of inflammatory myelopoiesis is of great interest in this particular field of study. While hinted at with discussion of Helicobacter pylori, it is clear that direct manipulation of the immune response is an important part of bacterial function. To that end, we have made our linkage from microbiome to inflammatory myelopoiesis more explicit in both The microbiome and cancer as well as Immunotherapy efficacy sections. We also expand upon the concept of inflammatory myelopoiesis and discuss how these outcomes could be achieved, either by way of directly driving production of effector T cells, or by enacting development of gamma-delta T cells for potentially more specific efficacy with fewer adverse effects. 

Reviewer 2 Report

Specific comments:

  • There is short list of abbreviations used in the study but some of them didn’t appear, e.g. PD-L1 (line 21). I understand that many abbreviations are so common that they don’t need to be explained but some of them (like PD-L1 and others) are not so maybe they should be explained;
  • Line 53 – “… plays a pivotal component in …” - maybe it would be reasonable to replace “component” by e.g. “role”?
  • There is a mess in Chapter 2. In the second paragraph there is about colorectal cancer, in the next about lung, then gastric and then again about respirator ones – should be clarified (maybe the Chapter should be subdivided?);
  • Last paragraph in the Chapter 2 – Jin ‘s study is mentioned so proper ref should also appear;
  • Bacteria should be written in italics – should be corrected e.g. in whole Chapter 3;
  • First paragraph in Chapter 4.1. – “… several studies …… have been performed” – multiple references should be added;
  • Last paragraph in Chapter 4.1. – again Jin’s work is cited so proper reference should also appear in this area;
  • Table 1 – It is not possible to read the content of the table – must be improved!
  • At many pints English language should be clarified.

Author Response

Round 2 

Reviewer 1 Report

Excellent review, discussing in detail novel approaches to Microbiome-based Therapies in Immuno-Oncology.